# Genetic and Clinical Characterization of Patients with HNF1B-Related MODY in Croatia

**DOI:** 10.3390/jpm13071063

**Published:** 2023-06-28

**Authors:** Maja Baretić, Domagoj Caban, Jadranka Sertić

**Affiliations:** 1Division of Endocrinology and Diabetes, Department of Internal Medicine, University Hospital Centre Zagreb, Kišpatićeva 12, 10000 Zagreb, Croatia; 2School of Medicine, University of Zagreb, 10000 Zagreb, Croatia; 3Department of Laboratory Diagnostics, University Hospital Centre Zagreb, 10000 Zagreb, Croatia

**Keywords:** diabetes mellitus, MODY, HNF1B, genetic, phenotype

## Abstract

Background: Mutation of the gene encoding Hepatocyte Nuclear transcription Factor-1 Beta (HNF1B) causes a rare monogenetic subtype of Maturity-Onset Diabetes of the Young (MODY). HNF1B-related MODY results in the dysfunction of multiple organ systems. However, genetic analysis enables personalized medicine for patients and families. Aims: To understand the clinical characteristics and explore the gene mutations in Croatian patients. Methods: This was a retrospective observational study of individuals (and their relatives) who were, due to the clinical suspicion of MODY, referred to the Department of Laboratory Diagnostics at the University Hospital Centre Zagreb for genetic testing. Results: A total of 118 participants, 56% females, were screened. Seven patients (three females) from five families were identified to have HNF1B-related MODY. The median age at diagnosis was 31 (11–45) years, the median c-peptide was 0.8 (0.55–1.39) nmol/L, the median HbA1c was 9.1 (5.7–18.4)%, and the median BMI was 22.9 kg/m^2^ (17–24.6). Patients had a variety of clinical manifestations; kidney disease was not as frequent as liver lesions, neuropsychiatric symptoms, hyperlipidemia, hyperuricemia, and hypomagnesemia. We identified two new pathogenic mutations (c.1006C > G protein p.His336Asp on exon 4 and c.1373T > G p protein Val458Gly on exon 7). Conclusions: In a study involving Croatian patients, new genetic (two previously unknown mutations) and clinical (diverse range of clinical presentations) aspects of HNF1B-related MODY were found.

## 1. Introduction

Personalized medicine is a rapidly advancing field that aims to tailor medical treatments to an individual’s specific genetic makeup. One area where personalized medicine has shown significant promise is in the diagnosis and treatment of Maturity-Onset Diabetes of the Young (MODY), a monogenic form of diabetes. MODY is a heterogeneous group of monogenetic diseases caused by pancreatic β-cell dysfunction. MODY is difficult to diagnose because its symptoms can be the same as other types of diabetes, mostly mimicking type 1 and type 2 diabetes. It is characterized by the early onset of the disease, the autosomal dominant mode of inheritance, preserved endogenous insulin production, the absence of antibodies to pancreatic β-cells, and the absence of symptoms of insulin resistance with the progressive deterioration of β-cell function [1].

Mutation of the gene encoding Hepatocyte Nuclear transcription Factor-1 Beta (HNF1B) causes one of the MODY subtypes, an older classification known as MODY 5. The HNF1B gene encodes a transcription factor for the synthesis of proteins that are essential for the development of the pancreas and β-cells; also, it is involved in the organogenesis of the liver, kidney, genitourinary system, neural tube, and many other tissues [2]. HNF1B is very closely related to so many other metabolic diseases, i.e., it negatively regulates white adipocyte differentiation. Downregulation of HNF1b enhances adipocyte differentiation, while upregulation of HNF1b inhibits this process [3]. In addition, HNF1B is closely related to other obesity comorbidities (such as fatty liver disease).

HNF1B mutations affect multiple organ systems in the body and are associated with different phenotypes.

The prevalence of MODY varies across different populations. The highest prevalence of MODY would be expected in populations with a higher frequency of consanguineous marriages [1], but, due to the low rate of genetic testing in some regions, it may be underestimated due to underdiagnosis or misdiagnosis. MODY is estimated to account for approximately 2–5% of all diabetes cases [4]. HNF1B-related MODY is a rare form referring to only 1–2% of MODY diabetes cases. The most common subtypes are MODY caused by glucokinase gene mutations (GCK) and MODY caused by Nuclear transcription Factor-1 Alpha (HNF1A) gene mutations [5]. HNF1B-MODY may have a higher prevalence in certain populations and ethnic groups; some studies have suggested a higher prevalence in populations of European ancestry. Nevertheless, it could also be explained by the more frequent genetic testing performed in European countries [6,7].

Genetic testing plays a crucial role in the implementation of personalized medicine for MODY [8]. By analyzing an individual’s DNA, specific genetic mutations associated with different types of MODY will allow for an accurate diagnosis, prediction of disease progression, and response to specific treatments. HNF1B-related MODY can be classified into two subtypes based on the type of genetic change that causes the condition: gene deletion and gene mutation. In the case of HNF1B-related MODY caused by gene deletion, a portion of the HNF1B gene is missing, producing a shortened and ineffective version of the protein. HNF1B-related MODY caused by a gene mutation involves a change in the DNA sequence of the HNF1B gene that alters the structure or function of the protein. The first HNF1B mutation was reported in 1997, and since then there have been many new mutations identified. Still, not all mutations have confirmed biological relevance [9].

The aim of this study was to investigate and gain a deeper understanding of HNF1B-related MODY within the Croatian population by analyzing the clinical characteristics and gene mutations present in affected patients.

## 2. Participants and Methods

This was a retrospective observational study of individuals who were, due to the clinical suspicion of MODY, referred to the Department of Laboratory Diagnostics at the University Hospital Centre Zagreb for genetic testing from April 2019 to April 2023. The study included a total of 118 participants, 52 men (44%) and 66 women (56%), with a median age of 39 (6–81) years. The participants were patients followed by pediatric or adult diabetologists who met the entry criteria. The first criterion was having a diagnosis of diabetes mellitus according to the American Diabetes Association guidelines (in two separate test samples): HbA1c ≥ 6.5% (47 mmol/mol), fasting plasma glucose ≥ 7.0 mmol/L (126 mg/dL), 2 h plasma glucose ≥ 11.1 mmol/L (200 mg/dL) during an oral glucose tolerance test, or a patient having symptoms of hyperglycemia with a random plasma glucose level ≥ 11.1 mmol/L (200 mg/dL) [10]. The second criterion was based on the clinical suspicion of MODY diabetes; the patient had early-onset diabetes (<25 years old), but if family history strongly suggested an inheritance pattern (many members having diabetes), older patients (<45 years) were screened too; negative pancreatic autoantibodies, persistently detectable c-peptide, and/or a family history of diabetes in one parent and other first-degree relatives of that affected parent, persistently raised fasting blood glucose during and after pregnancy, extreme sensitivity to sulfonylurea, or having one of the extrapancreatic features associated with different subtypes of MODY. Exclusion criteria included the following: criterion of type 1 diabetes-like insulinopenia and positive autoimmunity, clinical signs of insulin resistance (acanthosis nigricans, increased abdominal circumference, and obesity), or signs of other types of diabetes such as diseases of the exocrine pancreas, drug-related, or other primary endocrinopathies. Additionally, family members of individuals with confirmed MODY were included in the study, regardless of the criteria, and older family members were screened.

The blood samples of study participants were referred for molecular genetic analysis to the Department of Laboratory Diagnostics at the University Hospital Centre Zagreb.

The clinical data of the patients, with several parameters related to the patient’s medical history and current status, were collected. These parameters included sex, age, age at diagnosis of diabetes, body mass index (BMI), family history of diabetes and kidney diseases, glucose profile, glucose tolerance test, glycated hemoglobin (HbA1c), fasting c-peptide, antibodies targeting β-cells, and medical history of therapy used for the treatment of diabetes (diet, oral hypoglycemics, or insulin). All participants (or their parents in the case of pediatric patients) signed informed consent in order to participate in the study. The study was carried out in accordance with all ethical norms according to the consent of the Ethics Committee of the University Hospital Centre Zagreb (ed. number: 02/21-JG).

We also used the HNF1B score on each particular patient that was calculated on 17 items, including antenatal discovery, family history, and organ involvement (kidney, pancreas, liver, and genital tract). The performance of the score was assessed by a ROC curve analysis in a 433-individual cohort containing 56 HNF1B-related MODY cases. The cutoff threshold for the negative predictive value to rule out HNF1B mutations in a suspected individual was 8 (sensitivity 98.2%, specificity 41.1%, and negative predictive value over 99%) [11].

A peripheral venous blood sample was taken from each participant in a container with anticoagulant K3EDTA (Greiner Bio-One, GmbH, Kremsmuenster, Austria). Genomic DNA was extracted from whole blood samples using a QIAamp DNA Blood Mini Kit (Qiagen, Qiagen gmbh Hilden, Germany) and quantified using NanoDrop Lite Spectrophotometer (ThermoFisher Scientific, SAD, Waltham, MA, USA), following the manufacturer’s protocol.

Gene HNF1B was sequenced using the Sanger sequencing method (BigDye Terminator v3.1 Cycle Sequencing Kit, Applied Biosystems) on an AB Genetic Analyzer 3130xl (Applied Biosystems, SAD, Waltham, MA, USA) following standard protocol.

Multiplex ligation-dependent probe amplification (MLPA) assays were carried out following the manufacturer’s protocol (MRC-Holland, Amsterdam, the Netherlands) using P241-E1 MODY Mix 1. Capillary electrophoresis was performed on an AB Genetic Analyzer 3130xl (Applied Biosystems, Waltham, MA, USA), and data were analyzed using a Coffalyser (MRC Holland, Amsterdam, The Netherlands). The clinical interpretation of the variants was based on ClinVar, the Leiden Open the Variation Database (LOVD), the Human Gene Mutation Database (HGMD), Varsome, and the American College of Medical Genetics (ACMG) recommendations. Additionally, variants were verified by the in silico online predicting algorithms: Mutation Taster, SIFT, and PROVEAN.

## 3. Results

Seven patients were identified to have HNF1B-related MODY, which accounts for 5.9% of the total patients screened. These seven patients (four males and three females) were descended from five families. In this group of patients, the onset of HNF1B-related MODY occurred from a young to middle age, with a median age at diagnosis of 31 (11–45) years. The median c-peptide at diagnosis was 0.8 (0.55–1.39) nmol/L, the median BMI at diagnosis was BMI 22.9 kg/m^2^ (17–24.6), and the median HbA1c at diagnosis was 9.1 (5.7–18.4)% or 75 (39–178) mmol/L.

In the clinical presentation, diabetes was present in 7/7 patients, a family history of diabetes in 6/7 patients, hepatic steatosis in 4/7 patients, hyperlipidemia in 4/7 patients, neuropsychiatric disease in 3/7 patients, hyperuricemia in 2/7 patients, hypomagnesemia in 2/7 patients, and kidney disease in 1/7 patients. Characteristics of the patients are presented in Table 1. The median HNF1B score for patients diagnosed with HNF1B-related MODY was 4 (2–6).

Molecular genetic analysis revealed that four patients had a gene mutation, while the remaining three had a deletion of the exon. The study identified two distinct mutations, one located on exon 7 (c.1373T > G p protein Val458Gly) and the other located on exon 4 (c.1006C > G protein p.His336Asp). Deletion and mutations occurred on the long (q) arm of chromosome 17.

## 4. Description of Cases

### 4.1. Genetic Change Mutation c.1373T > G p Protein Val458Gly on Exon 7

Family 1 The mutation c.1373T > G p protein Val458Gly was found in a 45-year-old male diabetic patient with a positive family history of diabetes and kidney disease. There is a probability that the patient had mild fasting hyperglycemia at least 10 years before the diagnosis, but there was no evident medical data regarding it. The patient’s mother and grandmother had diabetes that required insulin therapy, both were underweight with negative pancreatic antibodies. The patient’s uncle had polycystic kidney disease. At the time of diagnosis, the patient’s glycated hemoglobin (HbA1c) was 6.4% (46 mmol/mol), his c-peptide level was 0.8 nmol/L (normal fasting range for laboratory for adults 0.37–1.42 nmol/L), and he tested negative for islet cell antibodies. In addition to diabetes, the patient had severe hypertriglyceridemia that led to pancreatitis. The patient’s body weight was normal with a BMI of 23.4 kg/m^2^. At the time of our study, he was treated with an individually prescribed diabetes diet, while hyperlipidemia was treated with a combination of fibrates and statins. The HNF1B score for the patient was 2.

The genetic change mutation c.1373T > G p protein Val458Gly on exon 7 is a substitution of the nucleotide thymine (T) with guanine (G) at position 1373 of the DNA sequence in exon 7 of the HNF1B gene. This results in the substitution of the amino acid valine with glycine at position 458 of the protein sequence. Figure 1 shows the DNA sequence analysis of the HNF1B gene with the mutation.

Bioinformatics tools used to predict the potential impact of amino acid substitutions on protein structure and function predicted that this mutation would have a pathogenic effect [12]. The impact of this mutation is reported in the ClinVar database as a mutation of unknown significance, likely to be pathogenic [13].

### 4.2. Genetic Mutation c.1006C > G Protein p.His336Asp on Exon 4

The presence of the c.1373T > G p protein Val458Gly mutation was found in three patients from two families.

Family 2 The patient diagnosed with c.1006C > G protein p.His336Asp mutation was a young female. In her case, fasting hyperglycemia was found during pregnancy at the age of 31 with HbA1c 5.7% (39 mmol/mol) and c -peptide 0.68 nmol/L. During pregnancy, she was treated with insulin. At the time of this study, she was treated with a high-calorie and low-carbohydrate diet (her BMI was 17 kg/m^2^). Diabetes runs in the family both in the first and second-generation members, and no one is using insulin. The HNF1B score for this patient was 2.

Family 3 The patient with the previously mentioned mutation was a young male diagnosed with diabetes at the age of 17. He was admitted to the emergency room due to diabetic ketoacidosis; at diagnosis, his HbA1c was 12% (108 mmol/mol) and c-peptide was 0.95 nmol/L, and his BMI was 22.9 kg/m^2^. Negative islet autoantibodies pointed to the non-autoimmune etiology of diabetes. The patient has dyslexia and hepatic steatosis (liver function enzyme levels were two times the upper limits of normal). The HNF1B score for this patient was 4. Diabetes runs in many family members, i.e., the patient’s father was diagnosed with an HNF1B-related MODY mutation during the family screening. In his case, hyperglycemia was found at the age of 37 with an HbA1c of 11% (97 mmol/mol), he was initially treated with oral hypoglycemics (metformin and dipeptidyl peptidase 4 inhibitor), and after a few years, insulin was introduced in therapy. The father has also hyperlipidemia and hepatic steatosis. The HNF1B score for the father was 4.

The mutation c.1006C > G, p.His336Asp is placed in the hybridization domain of the HNF1B gene in exon 4 of the HNF1B gene. It involves a substitution of cytosine (C) with guanine (G) at position 1006 of the gene, resulting in a change from histidine (His) to aspartic acid (Asp) at position 336 of the protein. Figure 2 shows the DNA sequence analysis of the HNF1B gene with the mutation.

Based on the classification of the mutation in the ClinVar database, it is reported as a mutation of unknown significance, likely benign. It is important to note that the classification in the ClinVar database is based on the available evidence at the time and many changes are possible as new evidence emerges [12,13].

### 4.3. Genetic Changes with Heterozygous Deletion of Exon 1–9

Exon 1–9 was deleted in three patients from two families. Figure 3 shows the multiplex ligation-dependent probe amplification (MLPA) of a heterozygous deletion of the HNF1B gene exons 1–9.

Family 4 The female patient with HNF1B gene exon deletion had a negative family history of diabetes or kidney diseases. Her initial presentation of diabetes at the age of 19 was rapid and severe; she had diabetic ketoacidosis and an extremely high HbA1c level of 18.4% (178 mmol/L), c-peptide 0.55 nmol/L, negative islet antibodies, and her BMI was 19.8 kg/m^2^. At the time of this study, the patient was being treated with insulin. In addition, she was diagnosed with selective mutism during childhood. Additionally, significant hepatic lesions (unexplained, otherwise with detailed workup) in the form of hepatic steatosis with enzymes more than five times the upper limit of normal were diagnosed as well as hypomagnesemia, probably related to the HNF1B mutation. The HNF1B score for this patient was 4.

Family 5 In the young female patient diagnosed with the same exon deletion, several family members have diabetes. She was diagnosed with diabetes at 11 with HbA1c 7.2% (55 mmol/mol), c-peptide 1.39 nmol/L (normal fasting range for laboratory children 0.13–0.72 nmol/L), and negative islet antibodies. Her BMI was 24.6 kg/m^2^. At the time of the study, she was treated with oral hypoglycemic therapy (metformin). In her laboratory results, there was hyperuricemia, hyperlipidemia, and hypomagnesemia, also probably related to the HNF1B mutation. The patient had hepatic steatosis with enzymes more than four times the upper limit of normal. The HNF1B score for that patient was 6. During the family screening, the mutation was found in her father who developed diabetes at the age of 36 and was also being treated with oral therapy for diabetes (metformin and dipeptidyl peptidase 4 inhibitor). In the father’s medical history, there was left renal agenesis, paranoid schizophrenia, hyperuricemia, and hyperlipidemia. The HNF1B score for the father was 4.

## 5. Discussion

This retrospective observational study in Croatia of individuals diagnosed with HNF1B-related MODY confirms that, in addition to causing diabetes, HNF1B mutations can also lead to a variety of other clinical manifestations, depending on the specific mutation and gene expression. In previous observations, the most common health condition associated with HNF1B mutation were renal cysts and diabetes syndrome (development of cysts in the kidneys together with the development of diabetes) [14]. Consistent expressivity of renal disease and diabetes is not frequently seen in described cases in Croatia [15]. Most of our patients diagnosed with HNF1B-related MODY did not have renal disease (only one patient had renal agenesis and one family member that was not genetically tested had a history of renal cysts), and many had other traits of the mutation. Such an observation should be taken with caution since kidney damage is diagnosed sometimes (like diabetes) later in life and some of our patients were diagnosed at a young age. In addition, the HNF1B score with a cutoff threshold of 8 for the negative predictive value to rule out HNF1B mutations would be not of much help in our cohort since no one fulfilled the criterion for the screening. As the authors of the score mentioned, the score is suitable for young patients with isolated renal malformations and that was not the case in our patients [11].

HNF1B-related MODY has been also associated with a range of extrapancreatic manifestations not related to the genitourinary tract. Patients with HNF1B mutations have a higher risk of developing hepatic lesions, such as liver adenomas and focal nodular hyperplasia, which can lead to hepatic dysfunction and potentially even liver failure. In our cohort, liver damage, unexplained by other diseases, was seen more frequently (found in four patients) than renal pathology. In personalized medicine, the approach to a liver lesion in HNF1B-related MODY would be to understand the underlying genetic condition rather than investigate the cause of liver damage. Avoiding unnecessary invasive procedures such as a liver biopsy is indeed a valid consideration in personalized medicine, especially if the cause of liver damage is already known to be related to the underlying genetic condition.

In addition, individuals with HNF1B-related MODY have an increased risk of developing metabolic disorders such as hyperlipidemia, including a low high-density lipoprotein level and elevated triglyceride level. This was found in four patients of our cohort, with one of them having pancreatitis because of severe hypertriglyceridemia. The upregulation of HNF1b results in the inhibition of peroxisome proliferator-activated receptor γ (PPARγ) and its target gene expression [2]. PPARγ is a key transcription factor involved in adipocyte differentiation, lipid metabolism, glucose homeostasis, and inflammation. Its activation promotes adipogenesis, improves insulin sensitivity, and modulates inflammatory responses in adipose tissue. Therefore, the inhibitory effect of HNF1b on adipocyte differentiation may be mediated, at least in part, through the suppression of PPARγ expression and its downstream target genes.

Furthermore, there is evidence to suggest that HNF1B mutations are associated with an increased risk of certain neurodevelopmental disorders, including autism spectrum disorder, intellectual disability, and learning disabilities. Of the seven total patients diagnosed with HNF1B-related MODY in this cohort, three of them had neuropsychiatric traits of the disease (one had dyslexia, one had autism, and one had paranoid schizophrenia). The exact mechanisms by which HNF1B mutations may contribute to these conditions are not yet fully understood and are the subject of ongoing research.

The HNF1B score (calculated upon 17 items including antenatal discovery, family history, and organ involvement) used as a tool to select patients for HNF1B gene analysis, also recognized uric acid as a variable of intermediate specificity, and hypomagnesemia as a variable of high specificity [11]. In the meta-analysis including 61 eligible patients from 15 countries, hypomagnesemia was found in 92% of patients having HNF1B-related MODY [15]. In our cohort, both hyperuricemia and hypomagnesemia were found in two of seven patients. Still, it is unclear whether hyperuricemia is an initial feature of HNF1B-related chronic kidney disease or an independent feature [16,17].

It is important to note that, while these extrapancreatic manifestations are associated with HNF1B-related MODY, not all patients with HNF1B mutations will develop them, and the severity and manifestation of these conditions can vary widely among affected individuals [18]. Those diverse cases point to the fact that HNF1B-related MODY is a rare form of diabetes, and its diagnosis requires a high degree of suspicion based on clinical presentation and family history. Even more, family history is not a mandatory feature, as in one of our patients. Sometimes, there is only one patient in a family, since de novo mutations account for about 50% of all variants, meaning family history may be missing during diagnosis [19].

The identification of two distinct genetic mutations linked to HNF1B-related MODY is an important finding because it sheds light on the underlying genetic mechanisms that contribute to this form of diabetes. These mutations were previously considered to be of unknown significance, and their effect on the HNF1B gene and its protein was not well understood.

The first mutation found was the c.1373T > G p protein Val458Gly on exon 7. The pathogenicity prediction tools found the mutation to probably be harmful since the location of the mutation is within the hybridization domain of the HNF1B protein. Such mutations affect the protein’s ability to bind to DNA, leading to altered gene expression and disruption of normal cellular processes. Based on the patient history of diabetes and hyperlipidemia and positive family history of diabetes and kidney disease, it appears that this mutation in the HNF1B gene is a pathogenic one associated with HNF1B-related MODY.

The second mutation is the c.1006C>G protein p.His336Asp on exon 4. The location of the mutation is predicted to be pathogenic by bioinformatics tools; the site is positioned in a highly conserved functional domain of the HNF1B protein [9]. Mutations in this domain are likely to disrupt the normal function of the HNF1B protein, leading to altered gene expression and disruption of normal cellular processes. The pathogenicity also confirms the presence of the mutation in two affected family members from Croatia, as well as the coexistence of other extrapancreatic traits associated with mutations of the HNF1B gene (hyperlipidemia and hepatic lesion, dyslexia). Moreover, such a mutation was found recently in two unrelated patients with diabetes in Siberia [20].

Genetic counseling plays a crucial role in the management of HNF1B-related MODY. It begins with confirming the diagnosis through appropriate testing and an assessment of the family history of diabetes to determine the likelihood of MODY being inherited (information about affected family members, their ages of onset, and any known genetic testing results). For the patients and their families, it is important to understand the inheritance pattern of MODY diabetes, which is typically autosomal dominant with a 50% chance of the mutation being passed on to each child of an affected individual. Observing our cases, it is clear that there are different phenotypes of the disease with consequences on different organ systems. Some appear at an early age, some later, some are barely noticeable in laboratory tests, and some have a severe clinical presentation. The task of a genetic counselor is to discuss the risks and implications of having HNF1B-related MODY diabetes, not only regarding the likelihood of diabetes complications and treatment options but also the possibility of it presenting neuropsychiatric heredity and how the disease impacts other family members. If the individual is of reproductive age and considering having children, the genetic counselor should provide information on family planning options to help individuals make informed decisions about their healthcare and family planning options.

Molecular genetic testing for the four most frequent MODY types was introduced in Croatia in 2017. Until now, scientific interest was mostly focused on HNF1A-related MODY and GCK-related MODY [21,22]. Since then, by identifying the underlying genetic mutation, precise diagnoses have been delivered, treatment plans tailored, and personalized prognostic information provided for many patients and families. Now, HNF1B-related MODY is recognized as a new challenge. As seen in the seven described cases, personalized medicine through HNF1B-related MODY genetic testing holds great potential for improving patient care. As our understanding of HNF1B-related MODY genetics continues to expand, the integration of genetic testing into clinical practice and personalized medicine will enhance the management of this complex disease.

## 6. Conclusions

It is interesting to note that, in this study of seven patients from Croatia diagnosed with HNF1B-related MODY, a wide range of clinical manifestations were observed. While HNF1B mutations are primarily associated with kidney disease and diabetes, it is crucial to consider the additional clinical features found more frequently in this cohort such as liver lesions, neuropsychiatric symptoms, and metabolic disturbances (hyperlipidemia, hyperuricemia, and hypomagnesemia) to fully understand the spectrum of this condition. The diverse clinical manifestations observed in the study underline the importance of considering a broader range of features when diagnosing and managing patients with HNF1B-related MODY. Recognizing and evaluating liver lesions, neuropsychiatric symptoms, and metabolic disturbances alongside kidney disease and diabetes can contribute to a more comprehensive understanding of the condition and facilitate personalized medical interventions for affected individuals.

We also identified two new pathogenic mutations (c.1006C > G protein p.His336Asp on exon 4 and c.1373T > G p protein Val458Gly on exon 7) and linked them to HNF1B-related MODY. The identification of those mutations highlights the importance of ongoing research into the genetic causes of diabetes, as it may uncover previously unknown mutations that contribute to the disease.

Overall, this study contributes to the body of knowledge surrounding HNF1B-related MODY in the Croatian population. It is worth mentioning that individual patient characteristics and variations in HNF1B mutations may contribute to the variability in clinical presentations. However, further research is necessary to elucidate the underlying mechanisms and establish clearer genotype–phenotype correlations in HNF1B-related MODY.

## Figures and Tables

**Figure 1 jpm-13-01063-f001:**
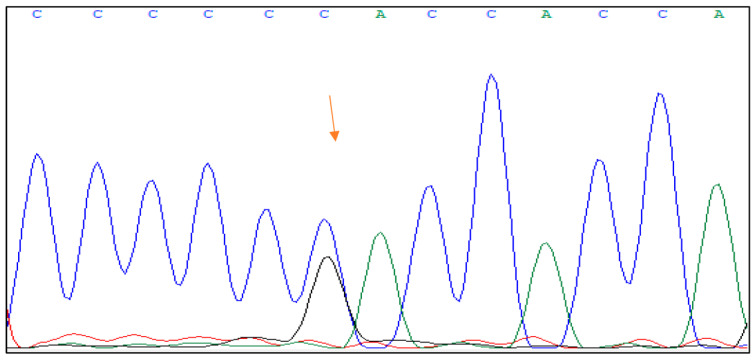
The Sanger sequencing of heterozygous mutation of HNF1B gene in exon 4 (c.1006C > G, p.His336Asp).

**Figure 2 jpm-13-01063-f002:**
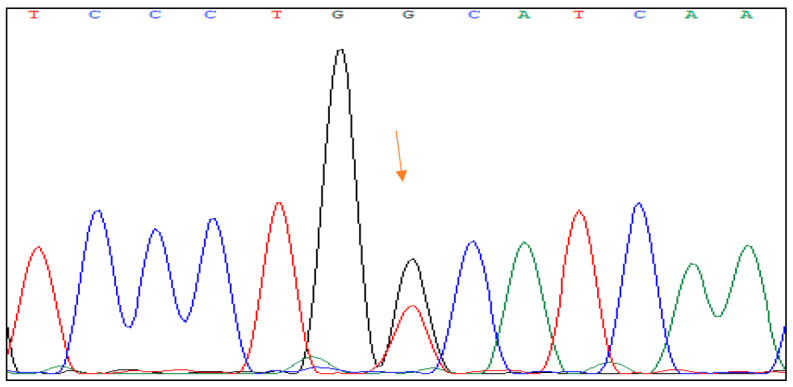
The Sanger sequencing of heterozygous mutation of HNF1B gene in exon 7 (c.1373T > G, p.Val458Gly).

**Figure 3 jpm-13-01063-f003:**
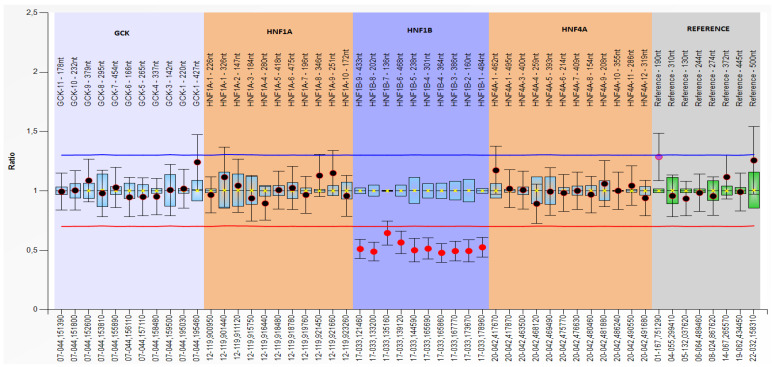
Multiplex ligation-dependent probe amplification (MLPA) of a heterozygous deletion of the HNF1B gene exons 1–9.

**Table 1 jpm-13-01063-t001:** Characteristics of patients diagnosed with HNF1B -related MODY.

Families	Patients	Genetics	Family History	Gender/Male or Female	Age at Diagnosis/years	BMI at Diagnosis/kg/m^2^	c-Peptide at Diagnosis/ nmol/L	A1C at Diagnosis/%	Kidney Disease	Liver Lesion	Neuropsych Disease	High Lipids	High Uric Acid	Low Mg
Family 1		Mutation 1												
	Patient 1		Yes	Male	45	23.4	0.8	6.4	No	No	No	Yes	No	No
Family 2		Mutation 2												
	Patient 1		Yes	Female	31	17	0.68	5.7	No	No	No	No	No	No
Family 3		Mutation 2												
	Patient 1		Yes	Male	17	22.9	0.95	12	No	Yes	Yes	No	No	No
	Patient 2	Father of Patient 1	Yes	Male	37	No data	No data	11	No	Yes	No	Yes	No	No
Family 4		Exon deletion												
	Patient 1		No	Female	19	19.8	0.55	18.4	No	Yes	Yes	No	No	Yes
Family 5		Exon deletion												
	Patient 1		Yes	Female	11	24.6	1.39	7.2	No	Yes	No	Yes	Yes	Yes
	Patient 2	Father of Patient 1	Yes	Male	36	No data	No data	No data	Yes	No	Yes	Yes	Yes	No

Legend Mutation 1—Genetic change is mutation c.1373T > G p protein Val458Gly on exon 7, Mutation 2—Genetic change is mutation c.1006C > G protein p.His336Asp on exon 4, Exon deletion—Genetic change is the heterozygous deletion of exon 1–9 HbA1c—glycated hemoglobin BMI—body mass index Mg—magnesium.

## Data Availability

The data presented in this study are available on request from the corresponding author.

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
