# Peer review of "Genetic and Clinical Characterization of Patients with HNF1B-Related MODY in Croatia"

_jpm, 2023, doi:10.3390/jpm13071063_

Round 1

Reviewer 1 Report

This reviewer thanks the authors for bringing this  inquisitive topic to the audience. The manuscript and the genetic mutations there in 7 participants are well described. I have the following two concerns and must have been mean while reviewing this manuscripts. They are:

1) in the results section a real data besides the descriptive style of the DNA sequencing results of the 7 patients, their important mutation parts (sections) has to be captures and place side by side to realize and or visualize the changes must be put in as figure to authenticate the results stated therein.

2) and HNF1B is very closely related to so many other metabolic disease such as fatty liver disease and or other obesity comorbidities, they are very much co-related and has to be highlighted. However looks very healthy in the Liver issues in those patients.

3) BMI as one co-variates in the table is strongly recommended.

4) Any molecular pathways can be think of. A good story could be in the discussion.

Author Response

Dear reviewers, dear Editor,

I wanted to express my gratitude for reviewing our manuscript titled "Genetic and Clinical Characterization of Patients with HNF1B-related MODY in Croatia." Your suggestions are greatly appreciated. Thank you for the opportunity to contribute to the scientific community.

Reviewer 1

  • in the results section a real data besides the descriptive style of the DNA sequencing results of the 7 patients, their important mutation parts (sections) has to be captures and place side by side to realize and or visualize the changes must be put in as figure to authenticate the results stated therein.

The figure of Sanger sequencing of heterozygous mutation of HNF1B gene of both mutations is added and multiplex ligation-dependent probe amplification (MLPA) of a heterozygous deletion of the HNF1B gene exons 1-9.

  • and HNF1B is very closely related to so many other metabolic disease such as fatty liver disease and or other obesity comorbidities, they are very much co-related and has to be highlighted. However looks very healthy in the Liver issues in those patients.

The part in introduction is added regarding the downregulation of HNF1b enhancing adipocyte differentiation

  • BMI as one co-variates in the table is strongly recommended.

BMI is added in the abstract, results and Table

  • Any molecular pathways can be thin of. A good story could be in the discussion.

The part of the upregulation of HNF1b resulting in the inhibition of peroxisome proliferator-activated receptor γ (PPARγ) and its target gene expression

Reviewer 2

  • The introduction section is fair and describes in a well-summarized way the MODY forms of diabetes and particularly the type 5 of the classic previous classification. Please, introduce a reference at the end of the first paragraph to justify it. Please, state the arm and chromosome where mutations or deletions of the HNFB1B are found.

Reference is added

  • Regarding the methodology, authors should clarify (according to ADA criteria) if the subjects meeting the three first DM criteria were diagnosed because of unequivocal hyperglycaemia according to two abnormal test results from the same sample or in two separate test samples.

It was diagnosed from two separate samples and it is added to the text

  • As of the Results section, it would be important to describe the hepatic lesions found in each case, as adenomas, focal nodular hyperplasia or hepatic steatosis have been previously described in these cases. Moreover, a “mild hepatic lesion” is meaningless for a physician.

Hepatic steatosis is described in more detail, mild is described in a way how many times liver enzymes were elevated.

  • Please, clarify the oral treatments of the patients from families 3 and 5.  

The treatment (the type of drugs) is described in those cases

  • English style and syntaxis is good but should be revised

English is changed according to the suggestions.

If there are any further suggestions, do not hesitate to contact me

Regards

Maja Baretić

Reviewer 2 Report

The paper under the title “ Genetic and Clinical Characterization of Patients with HNF1B-2 related MODY in Croatia” is presented by Maja Baretić, Domagoj Caban and Jadranka Sertić, from the Division of Endocrinology and Diabetes (Department of Internal Medicine, University Hospital Centre in Zagreb), the School of Medicine University of Zagreb in Zagreb, and the Department of Laboratory Diagnostics at University Hospital Centre in Zagreb (Croatia).

  Authors present a descriptive pheno and genotype analysis on HNF1B related MODY form of diabetes within the Croatian population by describing the clinical characteristics and gene mutations presented in several affected patients.

  Overall, the study is well presented and interesting. Sections are well introduced and described. The English style is good, although there are some few errors that should be corrected (see minor comments). The description of the two new pathogenic mutations of the HNF1B-related MODY is the most important finding of this study.

  Otherwise, authors should address some few issues prior to the publication of this study that I address subsequently.

  The introduction section is fair and describes in a well-summarized way the MODY forms of diabetes and particularly the type 5 of the classic previous classification. Please, introduce a reference at the end of the first paragraph to justify it. Please, state the arm and chromosome where mutations or deletions of the HNFB1B are found.

  Regarding the methodology, authors should clarify (according to ADA criteria) if the subjects meeting the three first DM criteria were diagnosed because of unequivocal hyperglycaemia according to two abnormal test results from the same sample or in two separate test samples.

  As of the Results section, it would be important to describe the hepatic lesions found in each case, as adenomas, focal nodular hyperplasia or hepatic steatosis have been previously described in these cases. Moreover, a “mild hepatic lesion” is meaningless for a physician.

  Please, clarify the oral treatments of the patients from families 3 and 5.  

 On the other hand, it would be interesting to present an easy descriptive table or figure with the clinical disorders found in each of these patients (hepatic disease, kidney disease, dyslipidaemia, neuropsychiatric, hyperuricemia, hypomagnesemia). It could be introduced at the end of the second paragraph of the Discussion section.  

 Minor comments

    English should be revised. Authors should correct some sentences and minor errors as follows:

 Change A1C to HbA1c across the draft (abstract and main body).

  Line 54:  “HNF1B-related MODY has been found to have a higher prevalence in certain populations and ethnic groups;” Please rewrite.

  Line 74: “Diagnos-tics” Please correct.

  Line 83 “:” instead of “;”.

  Line 84: “…but if family his-tory strongly suggested an inheritance pattern (many members having diabetes) older patients (<45 years) were screened too, …”.

 Please correct this sentence as it is hardly understood.

 Line 88: “fas-ting”. Please correct.

 Line 89: “sulfonylu-rea”. Please correct.

 Line 155: “This results….” Correct to “This result….”.

 Line 192: please change the word “brutal” to other more scientifically correct term.

 Line 270: “Also such mutation is found….” Change to  “Also such mutation was found….”

 Line 272: “As seen in described cases…” Better: “As it is seen in the described cases…”.

English style and syntaxis is good but should be revised. Authors should correct some sentences and minor errors as discussed in my comments. 

Author Response

(The authors gave the same response as above.)
